# Enhanced Metal Coating Adhesion by Surface Modification of 3D Printed PEKKs

**Inwoo Baek** [1,2] , **Chul-Min Lim** [3] **, Kyoung Youl Park** [3] **and Bong Ki Ryu** [1,*]

1 School of Materials Science and Engineering, Pusan National University, Busan 46241, Korea; iwbaek@kims.re.kr
2 Department for 3D Printing Materials, Korea Institute of Materials Science, Changwon 51508, Korea
3 Defense Space Technology Center, Agency of Defense Development, Daejeon 34186, Korea; cmlim@add.re.kr (C.-M.L.); kypark@add.re.kr (K.Y.P.)
* Correspondence: bkryu@pusan.ac.kr; Tel.: +82-51-510-3200

**Abstract:** PEKK (polyether-ketone-ketone) polymer has been actively studied in applying electronic devices in satellites owing to its excellent light weight and thermal resistance. However, the limitation of metal coating to form on the PEKK surface is due to the high-volume resistivity and surface resistance. Here, we have investigated the correlations between the chemical treatment of the surface and adhesion strength between polymer–metal coating. Three-dimensional printed PEKK objects were manufactured and nickel was deposited on the surface by electroless plating. As the concentration of $H_2SO_4$ increased from 12.5 to 14.3 mol/L, the pore diameter showed a tendency to increase. However, as growing pore induced connecting each other, the pore size re-decreased from 15.1 to 18.0 mol/L. To control pore size and uniformity, we investigated the pore diameter of 3D printed PEKK as a function of treatment time and temperature. Uniform pores were observed at a temperature of 50 °C which were formed after 10 min and the average pore size was 0.28 μm. After $H_2SO_4$ swelling, samples were re-treated in the $KMnO_4$-$H_3PO_4$ etching system for the hydrophilic group. $KMnO_4$ broken C=C bonding and generated hydrophilic groups such as -COOH and -OH, the contact angle decreased from 64.7 to 51.1° compared with $H_2SO_4$ swelling. XPS survey spectra confirmed that not only breaking C=C bonding but also increasing hydrophilicity due to -OH, -C-, -SO3 and the catalyst absorption of Pd was improved. As a result of adhesive strength by ASTM D3359, compared with the $H_2SO_4$ swelling, the $KMnO_4$-$H_3PO_4$ etching system showed 5B which is the best result in standard test methods by adhesive tape test and peeling amount on the tape was less than 0.01%.

**Keywords:** super engineering plastic; polyether-ketone-ketone; 3D printing; electroless plating; adhesion force

## 1. Introduction

PEKK is one of the PAEK (polyaryl-ether-ketone) family group engineering thermoplastic in a harsh environment since that has high mechanical strength, chemical resistance and thermal stability [1]. PEKK can be produced by step-growth polymerization reaction as a semi-crystalline resin which has thermal properties where the glass transition ($T_g$) and melting temperatures ($T_m$) are 143 and 300 °C, respectively, and these temperatures are satisfied for end-use applications such as medical implants, energy storage, military engineering and aerospace etc. [2–4]. Especially, in the aerospace field PEKK can be applied to satellite antennas since it accounts for about 50% of the total weight, reduction in the antenna weight is essential for overall cost reduction. For example, in the case of RISAT-1 developed and launched in India in 2012, the SAR antenna of 870 kg has dimensions of $6.29 \times 2.09 \times 0.22$ m$^3$ three panels were installed out of the total mass of 1858 kg, which is close to 50% of the total weight [5]. Commonly, SAR and array antennas are widely used for Low Earth Orbital (LEO) satellites, but they have some disadvantages like high

power consumption, large volume and weight. Increasing antenna volume and weight of the satellite makes lower the storage efficiency of the projectile and increases the launch cost. For example, LEO average launch cost is about USD 10,000/kg, and if a medium-size satellite is assumed to be 1 ton, the total cost will be approached to USD 10 million. In case of industry trends, 'Space X' and 'One Web' are launched 227 kg and 150 kg of LEO satellite, respectively [6]. Therefore, the key-points of commercial space technology are lightweight and packing density.

Furthermore, 3D printing technology can reduce the manufacturing cost and improve the antenna performance by designing the complex antenna geometry using lightweight polymer. Three-dimensional printing technology is based on computer-aided design (CAD) systems with shape freedom by simulation capabilities, therefore it has the advantage of cost reduction and time saving. The technology is classified into seven categories by ISO/ASTM 52900:2021, which are material jetting, material extrusion, photo polymerization, binder jetting, sheet lamination, powder bed fusion and directed energy deposition [7]. Among the various 3D printing methods, fused filament fabrication (FFF) is one of the material extrusion techniques that are commercialized because it can be able to fabricate a larger structure. However, commercial FFF materials are not suitable for the space environment which represents a temperature up to 120 °C and $10^{-6}$ torr vacuum conditions due to the low heat resistance of the polymer, and impossible to apply the antenna structure from the low conductivity [8]. For example, a nanosat satellite is designed using PEEK FFF 3D printing technique by Rinaldi et al. with considerable thermal and outgassing performance and 0.19% as total mass loss which satisfies the NASA standard [9]. Nevertheless, there is a limitation of confirming the structural possibility, not for electronical materials through the provision of conductivity. Therefore, it is required to secure 3D printing technology for the space environment to overcome existing limitations, material process technology and optimal metal coating process to improve antenna characteristics. In order to improve antenna performance, it is essential to form a uniform plating layer by electroless plating. Compared to other techniques, such as physical vapor deposition (PVD), spray is more difficult to form a uniform layer for a complex structure [10].

In addition, PEKK electroless plating is difficult due to its high-volume resistivity ($10^{14}$ Ω) and surface resistance ($10^{15}$ Ω). A low uniformity plating layer deteriorates conductivity and becomes a starting point for peeling of the plating layer, deteriorating the performance of the antenna. To improve adhesion force, the chromic acid etching solution is usually applied for ABS, PC, etc. However, $Cr^{6+}$ in chromic acid causes a serious environmental problem due to strong oxidation, and the $KMnO_4$ etching system is studied for an environmentally friendly surface treatment solution [11–13].

Herein, we have investigated the correlations between surface modification through acid solutions and the adhesion force of metal coating on 3D printed PEKK. To control the pore size formed through surface modification, the anchoring effect was strengthened. Additionally, to generate a hydrophilic group to the surface, the adsorption of Pd as a reaction catalyst was increased, thereby improving the plating uniformity and adhesion strength of electroless plating. These results show that the surface treatment method can be applied to the 3D printing technology, the possibility of various applications is proposed.

## 2. Materials and Methods

### 2.1. PEKK 3D Printing

The schemes of the metal coating process on 3D printed objects treated by surface modification are depicted in Figure 1. FFF 3D printer (Fortus 450 MC, Stratasys, Eden Prairie, MN, USA) was applied to the manufacturing of the 3D structures. The filament used Antero 800 NA (Stratasys, Eden Prairie, MN, USA) which is a PEKK thermoplastic polymer melting temperature and the glass transition temperature was 300 °C and 151 °C, respectively. Three-dimensional STL files were designed using Autodesk 123D design program to 200 × 200 × 200 mm$^3$ and sliced by the Stratasys software program. More

detailed information regarding 3D printing was explained by correlations between 3D printing parameter and accuracy [8].

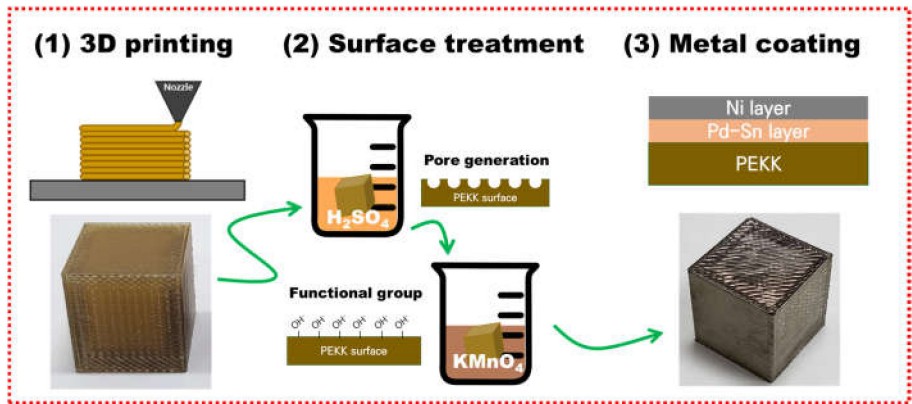

**Figure 1.** Illustration of metal coating on 3D printed PEKK.

## 2.2. Preparation of Electroless Plating Solutions

Electroless Ni-P plating was performed by a multi-step process shown in Table 1. Firstly, the 3D printed PEKK was degreased in alkaline solution at 60 °C for 15 min and rinsed with a large quantity of deionized (D.I) water 3 times. This step was performed to clean the surface due to oil grease and impurities. Second, PEKK samples were chemically swelled in 12.5, 13.4, 14.3, 15.1 and 18.0 mol/L sulfuric acid at room temperature for 3 min to control pore size and surface roughness. Third, the PEKK samples were etched in $KMnO_4$-$H_3PO_4$ system at room temperature for 20 min. Fourth, the etched PEKK samples were immersed in a KIAC conditioner (MSC cooperation, Incheon, Korea) for 10 min. Fifth, PEKK samples were neutralized 1.21 mol/L HCl and activated in catalysis bath at 35 °C for 10 min. In this process, PEKK surfaces were tightly attached to colloid Pd-Sn as a catalyst to initiate the electroless plating. After being rinsed with water, the sample were immersed in an activation bath in order to remove chlorine ions enveloping the Pd-Sn surface. Finally, activated PEKK samples were dipped into an electroless bath at 80 °C for 15 min. The pH of electroless bath was 5 which is adjusted using an ammonia solution. Except for the pre-dipping step, all samples should be rinsed with a large quantity of D.I water after each step.

**Table 1.** The composition of the solutions used in each step for electroless nickel plating.

| Step | Composition | Temperature, Time |
|---|---|---|
| Degreasing | NaOH 25 g/L, $Na_2CO_3$ 40 g/L, $Na_3PO_4$ 35 g/L | 60 °C, 15 min |
| Swelling | 12.5, 13.4, 14.3, 15.1, 18.0 mol/L $H_2SO_4$ | 25 °C, 3 min |
| Etching | $KMnO_4$ 38.5 g/L, 11.3 mol/L $H_3PO_4$ | 25 °C, 20 min |
| Conditioner | KIAC (MSC corp.) | 55 °C, 10 min |
| Neutralizers | 1.21 mol/L HCl | 25 °C, 1 min |
| Catalyst | $PdCl_2$ 0.8 g/L, SnCl2 50 g/L, 3.05 mol/L HCl | 35 °C, 10 min |
| Activation | 1.79 mol/L $H_2SO_4$ | 55 °C, 6 min |
| Electroless plating | $NiSO_4$ 25 g/L, $NaH_2PO_2$ 25 g/L, $C_6H_8O_7$ 25 g/L | 80 °C, 15 min |

## 2.3. Characteristic Evaluation

The surface topographies of the PEKK samples and Ni coating layer were observed by scanning electron microscopy (SEM, JSM-6610LV, JEOL Ltd., Tokyo, Japan). The pore size was measured by image analysis software (ImageJ version 1.53r). The surface composition and chemical bonding of samples were determined by X-ray photoelectron spectroscopy (XPS, Sigma Probe, Thermo Fisher Scientific, Waltham, MA, USA). The wettability of the

samples before and after pretreatment was evaluated by the static sessile drop method and analyzed by a drop shape analyzer (DSA100E, KRÜSS GmbH, Hamburg, Germany). The adhesion strength between the coating layer and substrate was tested by a qualitative scotch tape cross-cut test by ASTM D 3359 [14]. Adhesive tape was applied and removed over 16 cross hatched squares of $1 \times 1$ mm$^2$ made through the hatch cutter (Zehntner tester kit, Sissach, Switzerland).

## 3. Results and Discussion

### 3.1. Pore Generation by H$_2$SO$_4$ Swelling

The influences of H$_2$SO$_4$ concentrations were important to the etching step on surface properties on adhesion between polymer matrix and an electroless metal layer. In the etching step, H$_2$SO$_4$ reacted with PEKK carbonyl group and phenyl ring by protonation and sulfonation, respectively. PEKK was protonated by adding a proton H$^+$ to carbonyls in the main chain and sulfonated by replacing a hydrogen atom on phenyl ring with a sulfonic acid functional group [15]. Three-dimensional printed PEKK samples were etched by 12.5, 13.4, 14.3, 15.1 and 18.0 mol/L H$_2$SO$_4$, respectively.

As shown in Figure 2a, the surface morphology of raw PEKK was smooth and clear before etching. After 3 min of etching in 12.5 mol/L, Figure 2b shows that the surface was slightly roughened but the surface was too smooth and no pores were formed to the anchoring effect. With an increase in the solution concentration to 13.4 mol/L, some small cavities appeared on the PEKK surface, but the number of pores formed was too small and the depth of the cavities was shallow. When increased to 14.3 mol/L, Figure 2c shows that approximately 2~3 μm pores which led to an anchoring effect between PEKK and Ni layer appeared to surface and grew as the H$_2$SO$_4$ concentration increased [16]. However, the formed pores gradually grew at 15.1 mol/L concentration and met each other as shown in Figure 2d, the size of pores decreased and the anchoring effect could not work. This phenomenon was more severe at 18.0 mol/L concentration which was undiluted H$_2$SO$_4$, the PEKK surface is relatively smooth as if there was no surface treatment.

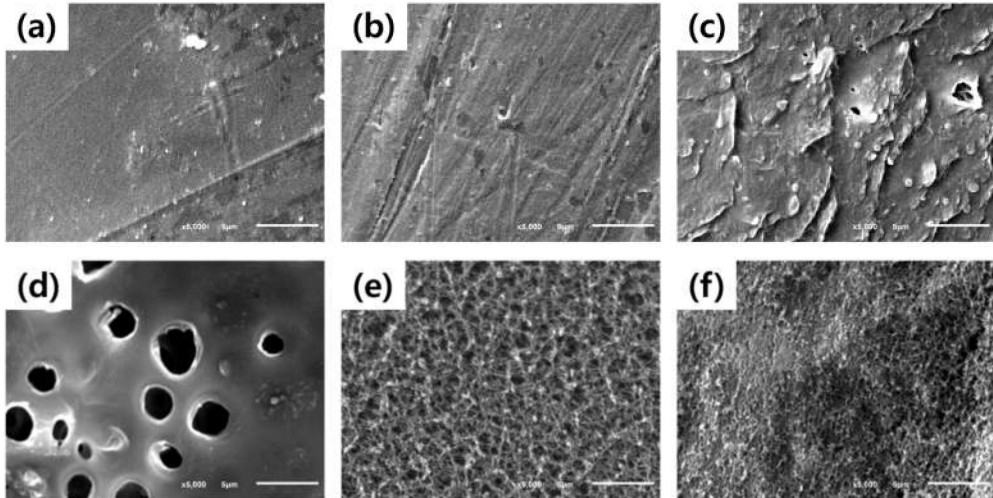

**Figure 2.** SEM images of PEKK surface after H$_2$SO$_4$ treatment: (**a**) Untreated PEKK, (**b**) 12.5 mol/L H$_2$SO$_4$, (**c**) 13.4 mol/L H$_2$SO$_4$, (**d**) 14.3 mol/L H$_2$SO$_4$, (**e**) 15.1 mol/L H$_2$SO4 and (**f**) 18.0 mol/L H$_2$SO$_4$ for 3 min at room temperature.

The effect of increased H$_2$SO$_4$ concentration etchings demonstrated differences in the alteration of PEEK surface. The difference between PEEK surface irregularities was due to the dissolution of the PEEK in H$_2$SO$_4$ by sulfonation reaction. Diluted H$_2$SO$_4$ from adding water into 18.0 mol/L leads to changes in chemical equilibrium in solutions and decreasing degree of sulfonation reaction. Therefore, the lower concentration of sulfuric acid did not create severe surface corrosion of the PEEK surface compared to the higher concentration.

Consequently, it could be derived that the etching process should be performed at a concentration of 13.4 mol/L $H_2SO_4$ which is the minimum concentration at which pores are formed, and the concentration was fixed at 13.4 mol/L to confirm the effect on temperature and time.

In order to derive the pore formation for the PEKK surface, the change in pore size was observed by changing the time and temperature at the sulfuric acid concentration of 13.4 M $H_2SO_4$. In Figure 3a, 25 °C swelling solution confirmed that the pores were generated but there were a few observed on the surface. At a temperature of 50 °C, pores were observed to form after 10 min and the average pore size was 0.28 μm in Figure 3b. It was confirmed that the anchoring effect, which had a positive effect on the adhesive strength, could be formed as uniform and tiny pores were formed on the surface. However, the temperature increased to 75 °C and the pores gradually grew as the time increased. After 3 min of solution treatment, the pores were observed at 0.43 μm and grew rapidly to 1.12 μm in 5 min, finally 1.46 μm in 10 min in Figure 3c. Since too large pores adversely affect the adhesive strength, it is necessary to control the optimal size through surface treatment.

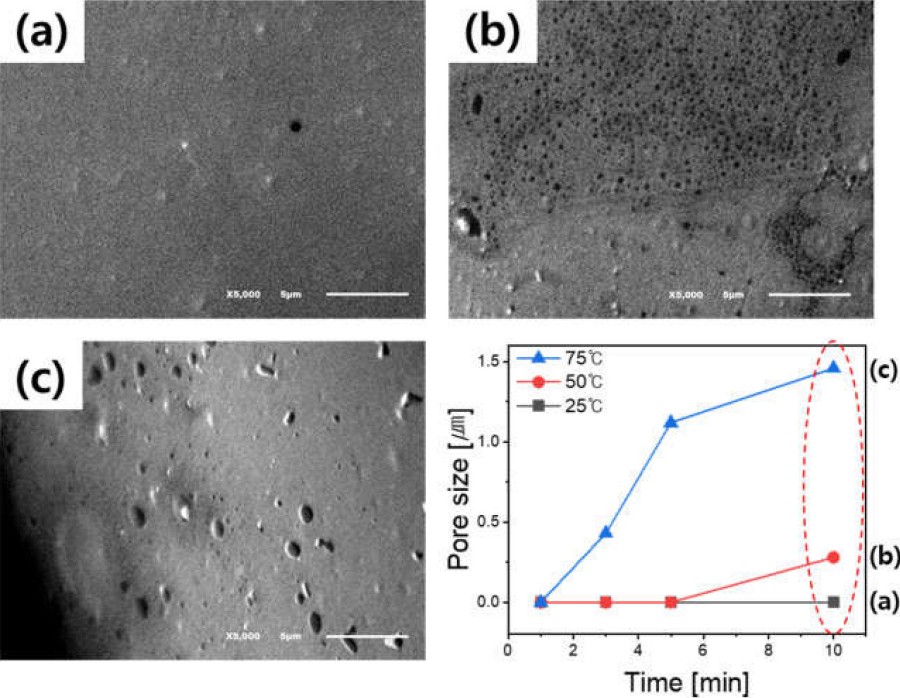

**Figure 3.** Pore diameter of 3D printed PEKK as a function of swelling time and temperature: (**a**) 25 °C, (**b**) 50 °C and (**c**) 75 °C.

### 3.2. Hydrophilicity Increase by KMnO$_4$-H$_3$PO$_4$ Etching

For electroless plating, the reaction of catalyst adsorption should be strongly configured. If the catalyst Pd was not strongly adsorbed on the PEKK surface, it would affect the operation and reliability of the electronic device. Although hydrophilic groups were formed on the surface through sulfonation and protonation reacting with sulfuric acid, they were not strong enough and required additional surface treatment. Therefore, in this study further etching was performed using the $KMnO_4$-$H_3PO_4$ etching system after $H_2SO_4$ swelling treatment.

The typical water droplets evaluated the contact angle of 3D printed PEKK samples as presented in Figure 4. For all the samples, hydrophilicity was increased after swelling and etching treatment. Compared with the untreated sample, it was increased significantly by $KMnO_4$ etching treatment. As shown in Figure 4a, the water contact angle of the untreated PEKK was 85.7°. After $H_2SO_4$ swelling treatment in Figure 4b, the water contact angle decreased to 64.7° due to -SO$_3$H and -OSO$_3$H generated by sulfonation and protonation

reaction, respectively. The hydrophilicity of the surface was further strengthened by $KMnO_4$-$H_3PO_4$ treatment, the water contact angle was decreased to 51.1° in Figure 4c. According to the previous studies, hydrophilicity was increased after swelling due to $KMnO_4$ broken C=C bonding at polymer structure and generated hydrophilic groups such as -COOH and -OH [12].

$$MnO_4^- + H^+ = MnO_2 + O_2 + H_2O \tag{1}$$

$$MnO_4^- + H^+ = Mn^{2+} + H_2O \tag{2}$$

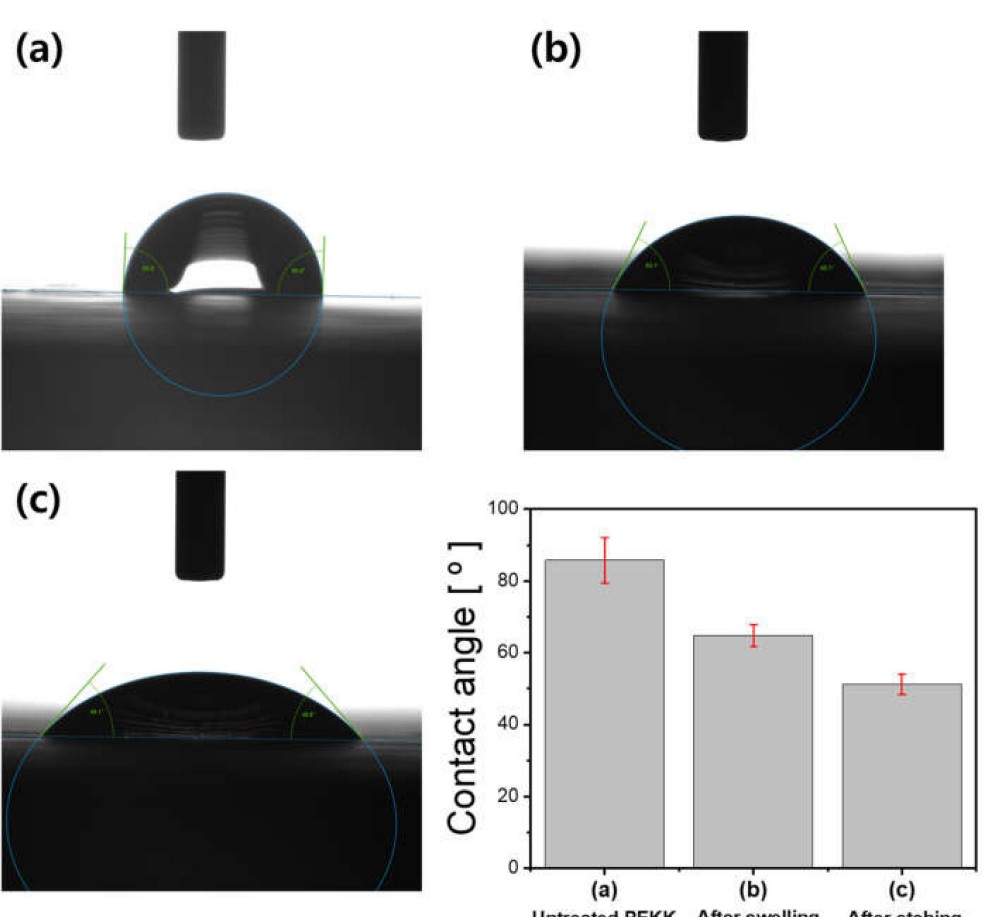

**Figure 4.** Contact angle measurement of a water drop on a 3D printed PEKK surface for comparing chemical treatment: (**a**) Untreated PEKK; (**b**) After $H_2SO_4$ swelling; (**c**) After $KMnO_4$ etching.

XPS spectra were performed to analyze the change in surface properties of the PEKK surface before and after treatment. Figure 5a shows XPS results which are before and after chemical treatment of the PEKK surface. Results shows that the carbon content signal peak decreased from 83.5% to 67.6% after surface treatment through sulfonation and protonation reactions, while the oxygen content increased from 15.6% to 27.4%. This phenomenon can be explained during the swelling process, new hydrophilic groups such as -OH and -COOH were generated from C=C of PEKK was destroyed and reacted by dehydration of $H_2SO_4$. In high-resolution C1 spectra for changes in surface functional groups, the PEKK subpeak consists of four types, with a C-C group (phenyl) at a binding energy of 284.61 eV, a C-O group (ehter) at 286.23 eV, C=O (carbonyl) at 288.46 eV and PEKK satellite peaks at 291.69 eV [12].

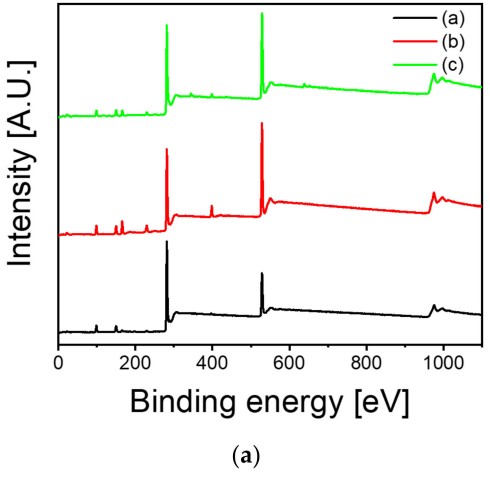

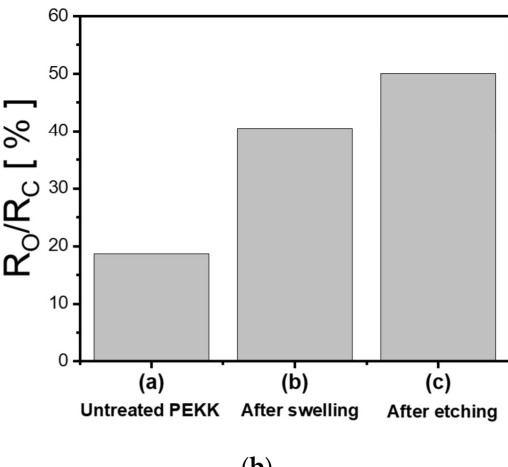

**Figure 5.** XPS survey spectra and carbon–oxygen content ratios of 3D printed PEKK samples: (**a**) XRD survey spectra; (**b**) Ratio of C1s and O1s peak for the prepared samples.

Additionally, $KMnO_4$-$H_3PO_4$ etching was oxidized by a strong oxidizing agent of $KMnO_4$, C content and O content ratios were changed to 52.3% and 31.2% compared to the surface treatment with sulfuric acid, the ratio of carbon to oxygen was finally increased to 50.08 in Figure 5b and Table 2. Through this, it was confirmed that -C- as well as -COOH was formed during surface treatment with the $KMnO_4$ system, and the formed -OH, -C-, -$SO_3$ and -COOH complexly increased the hydrophilicity of the surface, the catalyst adsorption of Pd was improved.

**Table 2.** Surface composition and the ratio of carbon–oxygen contents: (a) Untreated PEKK; (b) After $H_2SO_4$ swelling; (c) After $KMnO_4$ etching.

| Sample | C (%) | O (%) | Mn (%) | S (%) | P (%) | O/C |
|---|---|---|---|---|---|---|
| (a) Untreated PEKK | 83.5 | 15.6 | 0.1 | 0.7 | 0.1 | 18.70 |
| (b) After $H_2SO_4$ swelling | 67.6 | 27.4 | 0.1 | 4.8 | 0.1 | 40.50 |
| (c) After $KMnO_4$ etching | 52.3 | 31.2 | 3.6 | 2.8 | 0.1 | 50.08 |

*3.3. Adhesion Strength between PEKK-Ni Coating*

When the uniform and tiny pores are formed, this will enhance adhesion strength due to the anchoring effect between PEKK and electroless Ni coating. Figure 6 shows the ASTM D3359 cross-cut adhesion test and investigates the change in the adhesion strength with different surface modifications. In the case of surface treatment using sulfuric acid from Figure 6a, about 29.56% of the total area was peeled off indicating relatively poor results of grade 2B. However, when the surface was treated through the $KMnO_4$-$H_3PO_4$ system as seen in Figure 6b, about 0.01% of the area was peeled off, and it was confirmed that the adhesive strength was improved with a result of 5B, the highest grade. Therefore, a nickel electroless plating layer with uniform and excellent adhesion was formed on the surface through the result of plating adhesion strength for PEKK electronic parts with low voltage loss.

Compared with the reference electroless plating, we verified $KMnO_4$-$H_3PO_4$ etching solution through ASTM D3359 standard. In Table 3, four different research cases were used: acrylonitrile butadiene styrene (ABS), polyethylene terephthalate (PET), epoxy resin and polyimide (PI), respectively. According to Akin et al., $CrO_3$ was used as a strong oxidizing agent to form the Cu layer. However, copper was removed by adhesive tape and peeled off along the square grid as a result of the tape adhesion test [17]. An additional result from Jung et al. confirmed the importance of the etching solution, highlighting that it affected adhesion strength due to weak oxidizing power [18]. Third, Anderetta et al. reported that

the PAA solution showed feasibility for epoxy resin, but there was the most peeled metal observation along the grid lines among these studies [19]. Lastly, Ren et al. have suggested the KNaC$_4$H$_4$O$_6$ solution for electroless plating on PI due to its verified excellent adhesive strength of 5B [20].

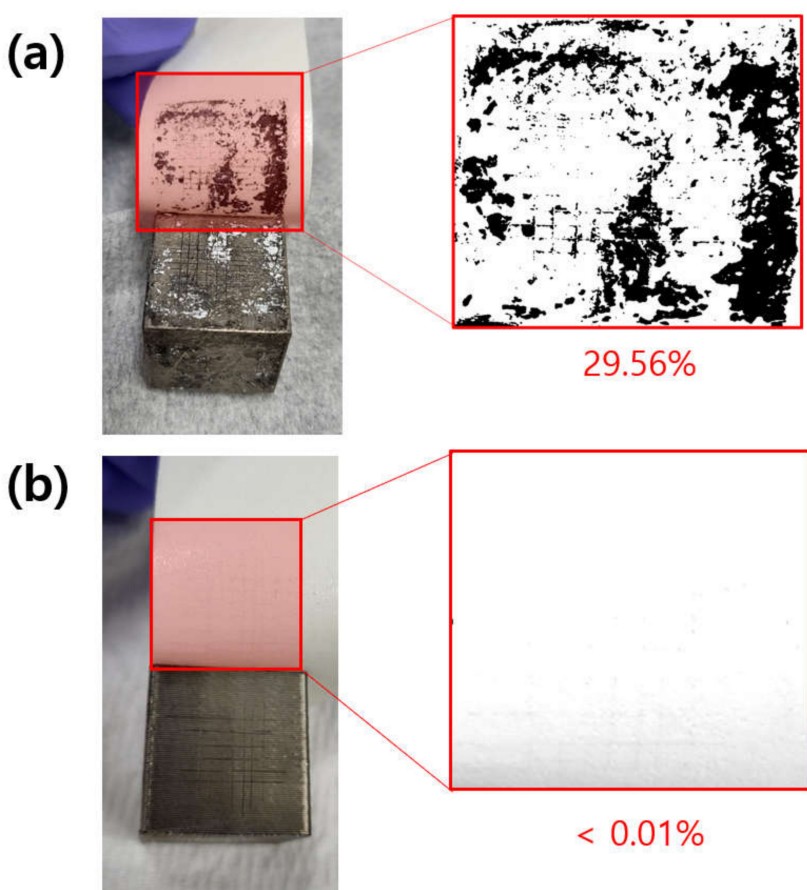

**Figure 6.** Images of ASTM D3359 adhesive strength results: (**a**) After H$_2$SO$_4$ swelling; (**b**) After KMnO$_4$ etching.

**Table 3.** Comparisons of reference electroless plating results.

| Reference | Substrate | Solution | Result | Before Adhesion Test | After Adhesion Test |
|---|---|---|---|---|---|
| Akin et al. [17] | ABS | CrO$_3$ | 4B or 5B | | |
| Jung et al. [18] | PET | HCl | 5B | | |
| Andretta et al. [19] | Epoxy | PAA | 4B | | |
| Ren et al. [20] | PI | KNaC$_4$H$_4$O$_6$ | 5B | | |

## 4. Conclusions

In this study, the correlation between chemical treatment and adhesion strength between polymer–metal affected by surface modification was investigated. The PEKK surface activated by sulfuric acid affected the concentration of sulfonation and protonation reaction. Pores started to generate on the PEKK surface and approximately 2–3 μm pores were observed at 14.3 mol/L by SEM image. First, uniformly formed pores had a positive effect on the adhesive strength through the anchoring effect and controlling the pore size was studied. In order to precisely control the pore size of PEKK surface, it was important to find the correlations between solution time and temperature. At 13.4 mol/L, which was the starting point of pore generation, the potential for controlling the diameter and uniformity of pores was experimentally shown according to the solution temperature. The average pore diameter of the PEKK surface was in proportion to the solution temperatures between 25 to 75 °C, increased from 0.28 to 1.46 which was associated with the anchoring effect. Second, we added $KMnO_4$-$H_3PO_4$, another surface treatment solution for enhancing hydrophilicity and functional group after $H_2SO_4$ treatment. Compared with $H_2SO_4$ treatment, the contact angle decreased from 64.7 to 51.1 °C due to C=C oxidation and creation hydrophilic groups such as -COOH, -OH by $KMnO_4$ which were strong oxidizing agents. This phenomenon could be explained by XPS spectra, the greater carbon and oxygen rate of the PEKK surface. The O/C ratio of PEKK samples increased from 18.70 to 40.50 according to $H_2SO_4$ swelling, and then increased up to 50.08 following $KMnO_4$-$H_3PO_4$ etching treatment. Third, the adhesion force between the PEKK-Ni coating was analyzed by a ASTM D3359 cross-cut adhesion test and we investigated the change in the adhesion strength with different surface modifications. Furthermore, compared to other studies, we confirmed the enhanced adhesion strength from $KMnO_4$-$H_3PO_4$ effects and the highest value of 5B by ASTM D3359. Consequently, this research will lead to not only space antenna but also electric part applications such as spaceship, drone and personal air vehicle (PAV).

**Author Contributions:** Conceptualization, K.Y.P.; methodology, I.B.; data curation, I.B. and C.-M.L.; writing—original draft preparation, I.B.; writing—review and editing, B.K.R.; supervision, B.K.R.; project administration, K.Y.P.; funding acquisition, C.-M.L. All authors have read and agreed to the published version of the manuscript.

**Funding:** This research was funded by Agency of Defense Development, Republic of Korea. (grant number: UD210003ED).

**Institutional Review Board Statement:** Not applicable.

**Informed Consent Statement:** Not applicable.

**Data Availability Statement:** All data have been included in the paper.

**Conflicts of Interest:** The authors declare no conflict of interest.

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
