# Peer review of "Enhanced Metal Coating Adhesion by Surface Modification of 3D Printed PEKKs"

_coatings, doi:10.3390/coatings12060854_

Round 1

Reviewer 1 Report

A fascinating study, which attracts me a lot. I just have some minor questions for concern.

  1. In the introduction part, please add a paragraph to show the importance of the study.
  2. is fig 2 is in situ SEM test? or just different samples.
  3. How to determine the pore size in fig 3.
  4. The results are impressive, please show it point by point.

Author Response

Thank you for your kind advice and review. Please see the attachment.

Reviewer 2 Report

" Enhanced Metal Coating Adhesion by Surface Modification of 3D Printed PEKK"

It is very interesting to focus to investigate the correlations between the chemical treatment of the surface and adhesion strength between polymer-metal coating. However, there are a few corrections that are essential to meet the standard for publication. Please refer to the following comments.

1) One of the features of this research is the use of 3D printed PEKK. Learn more about the PEKK3D prints used in this study and the originality of this study in the Materials and Methods section.

I feel that your research lacks distinctive features in the field of 3D printing, so please add more explanation.

2) The figure in the explanation of Figure 4. is difficult to understand. Please change the explanation and the figure in an easy-to-understand manner.

3) Your research will lead to the development of materials that will be useful in the future. Please add to the discussion section about the prospects for the actual use of this study.

Author Response

(The authors gave the same response as above.)

Round 2

Reviewer 2 Report

Thank you for giving me this opportunity to re-review your revised manuscript. 

I am happy that all of the suggested corrections have been made.

Thank you for spending so much time for revised manuscript.